# Advanced 3D Models of Human Brain Tissue Using Neural Cell Lines: State-of-the-Art and Future Prospects

**DOI:** 10.3390/cells12081181

**Published:** 2023-04-18

**Authors:** Rachele Fabbri, Ludovica Cacopardo, Arti Ahluwalia, Chiara Magliaro

**Affiliations:** 1Research Center “E. Piaggio”, University of Pisa, Largo Lucio Lazzarino 1, 56122 Pisa, Italy; 2Department of Information Engineering (DII), University of Pisa, Via G. Caruso 16, 56122 Pisa, Italy; 3Interuniversity Center for the Promotion of 3R Principles in Teaching and Research (Centro 3R), Italy

**Keywords:** in vitro advanced brain models, neuroblastoma cell lines, glioblastoma cell lines

## Abstract

Human-relevant three-dimensional (3D) models of cerebral tissue can be invaluable tools to boost our understanding of the cellular mechanisms underlying brain pathophysiology. Nowadays, the accessibility, isolation and harvesting of human neural cells represents a bottleneck for obtaining reproducible and accurate models and gaining insights in the fields of oncology, neurodegenerative diseases and toxicology. In this scenario, given their low cost, ease of culture and reproducibility, neural cell lines constitute a key tool for developing usable and reliable models of the human brain. Here, we review the most recent advances in 3D constructs laden with neural cell lines, highlighting their advantages and limitations and their possible future applications.

## 1. Introduction

Despite considerable efforts, we still have a very limited understanding of how the brain works when it is healthy or sick. Non-invasive imaging methods performed in humans lack the spatial and temporal resolution to probe its microscopic anatomy and function. Thus, a simplified and accessible model of the human brain is urgently needed.

Although animal models have provided a significant boost to neuroscience research and are still widely used, their limits have been extensively demonstrated. In fact, besides the ethical issues, studies have shown that the results of animal experiments often fail to translate into human clinical trials [1]. Moreover, microscopic studies of post-mortem human brains have revealed neural structures, enhanced wiring, and forms of connectivity among nerve cells not found in other animals [2,3,4]. In this scenario, it is crucial to develop more reproducible models, exploiting human cells for facilitating the translatability of the results obtained to humans [5]. Transitioning to non-animal models is also in compliance with the 3R (Replacement, Reduction and Refinement) principles [6].

Human neural cells cultured in highly controllable and monitorable environments have been widely used for investigating neurotoxicity, neuroprotection, drug screening and therapeutic assessment for both brain tumors and neurodegenerative diseases, e.g., Parkinson’s and Alzheimer’s [7,8,9]. Possible sources of human cells are primary neurons, immortalized neural stem cells from embryonic stem cells and cell lines. Adult neural stem or progenitor cells have also been used to investigate neurodegenerative and neurological disorders, brain cancer and ischemia in vitro [10,11,12]. However, primary neurons from human brains are often prohibitively expensive, mainly because of a lack of accessibility and the costs associated with harvesting and isolation. Indeed, in many countries, brain tissue is practically unobtainable because of cultural issues [13], and there are challenges regarding where it can be collected, obtaining ethical approval and access to donors. Moreover, inter-individual variability among donors limits the standardization of procedures for their characterization and culture. Furthermore, primary neurons do not undergo cellular division, limiting the number of experiments that can be performed [14]. On the other hand, embryonic or induced pluripotent stem cells can be differentiated into neural cells. Although these cells offer an unprecedented opportunity for investigating the pathogenesis of neurodegenerative disorders, culturing and maintaining stem cells is, again, highly expensive and technically challenging. Indeed, culture and environmental conditions can alter their capacity for self-renewal and differentiation [15]. As an alternative to primary or stem cells, researchers can also exploit established cell lines, which have important benefits. They have lower costs, can be cultured more easily than primary neurons and they can expand almost indefinitely. Hence, an (almost) unlimited number of cells are available, as long as they are not induced to differentiate, allowing for experiments with several duplicates and many different conditions [14]. Furthermore, cell lines are not beset with the ethical issues associated with culturing human primary neurons and stem cells or with experiments involving animals [16,17]. However, human neural cell lines often have malignant origins, whose genetic drifts may hamper their physiology and integrity [14].

Since soma are unrealistically flattened and neuronal axonal and dendritic outgrowth cannot occur in all directions in traditional monolayers [18], advanced models have been developed where cellular protrusions arranged in space are characterized by more physiological neural dynamics [19]. In particular, brain or cerebral organoids, which are derived from human (usually pluripotent) stem cells, can mimic the 3D (three-dimensional) structure and salient functional features of the brain [20,21,22]. However, despite their use for exploring developmental diseases and neurodegenerative disorders, brain organoids still suffer from the so-called ‘batch syndrome’ (variability from batch to batch), thus they lack reproducibility in generating cellular diversity and producing mature traits [23]. Moreover, their physiological relevance and translational potential is often hindered by non-viable cores, probably due to limitations in nutrient diffusion [23,24,25].

Three-dimensional constructs have also been generated from neural cell lines and may provide a strategy for developing standardized models with better physiological relevance compared with traditional 2D (two-dimensional) cultures. In this context, after an overview of the neural cell lines commonly used in the literature, we describe recent approaches exploiting cell lines for generating 3D models of brain tissue. Given the legislative and public urge to reduce the use of animals in scientific experiments, we suggest their re-evaluation in humane and human-relevant research, particularly for regulatory applications where standardized and reproducible inter-laboratory outcomes are crucial.

## 2. Search Methodology

To identify articles dealing with three-dimensional in vitro human-relevant models of brain tissue involving cell lines, we first conducted an analysis of the existing scientific literature. Web of Science was used with the query: (((neur* OR brain OR cereb*) AND model*) AND (neurosphere* OR 3d OR three-dimension*) AND human AND “cell* line*”). Only studies published from 2000 to 2022 were selected, thus identifying 304 original articles and 37 review papers.

Oncology (18%) and science technology (18%) are the research areas where cellular models of human brain tissue are mainly involved (Figure 1). Scientific efforts are mostly focused on improving cell culture methods (i.e., protocols for scaffold-based or scaffold-free spheroid cultures, novel technologies for monitoring cell parameters, imaging techniques suitable for cell cultures and genetic analyses of in vitro cells) and on applying such technologies to drug testing, the analysis of signaling pathways and subcellular mechanisms crucial in cancer development.

After this general inspection, a more precise analysis was carried out. Each abstract was read and assessed for inclusion according to whether its focus was on 3D models of brain tissue or neural tissue generated from immortalized human cell lines. Thus, a consistent number of papers was discarded because they focused on non-human models (e.g., murine, canine or porcine models), in vivo models, models developed with induced pluripotent stem cells and primary cells, or papers describing models of tissues different from the brain (e.g., colorectal, breast and pancreatic cancer models). Thus, a total of 96 papers (85 original articles and 11 review papers) was selected.

## 3. Cell Lines in 3D Culture

Most of the cell lines used in 3D in vitro models in neuroscience research derive from tumor tissues, in particular glioblastoma and neuroblastoma. In the following paragraphs, we describe the studies identified with the literature search in which 3D in vitro models of brain tissue have been developed using cell lines derived from tumors and from healthy tissues.

### 3.1. Cancer Cell Lines

The cancer cell lines most exploited in cellular neuroscience research derive from glioblastoma (52% of all papers) and neuroblastoma (35%). However, 3D models of brain tissue have also been generated using cells from embryonal carcinoma and medulloblastoma.

#### 3.1.1. Glioblastoma Cell Lines

Glioblastoma cell lines are used in 3D in vitro constructs for modeling gliomas, in particular glioblastoma multiforme (GBM), the most common malignant brain tumor with the poorest prognosis and survival [26,27]. Most of these 3D in vitro models are focused on the development of new therapies. However, cancer treatments and the possibility of unravelling cellular and subcellular mechanisms associated with cancer are also assessed by culturing 3D glioblastoma models. The most common glioblastoma cell lines used for 3D in vitro constructs are U-87MG [28], U-251MG, U-373MG [29,30], A 172 [28] and T-98G [31]. Table 1 summarizes their origin, gender, and the age of the sample from which the cell lines were derived, the cell morphology and the first ever citation, while a general overview of the applications employing glioblastoma cell lines in 3D in vitro models is reported in Table 2.

As detailed in Table 2, glioblastoma cell lines have been widely used for generating spheroids, even in co-cultures, mainly for oncological applications, e.g., for evaluating the effects of chemotherapeutic agents at the microscale [33,34] and for better characterizing the effects of microenvironment on cell invasive behavior [37]. However, attempts at exploiting such cells in 3D environments for toxicological applications are also present in the literature [43].

#### 3.1.2. Neuroblastoma Cell Lines

Neuroblastoma is the most common extracranial solid tumor observed in childhood (less than five years old) [44,45], originating from precursor cells of the neural crest. Table 3 summarizes the origin, gender and age of the sample from which cell lines were derived, as well as salient information about cell morphology and their first introduction, while Table 4 recaps the main findings for neuroblastoma-derived cell lines. Where 3D models are concerned, the most widely used neuroblastoma-derived cell line is the SH-SY5Y cell line (68%, versus IMR-32 (24%), HTLA-230 (4%) and Kelly (4%)).

As regards the applications, even for neuroblastoma cell lines, most of the efforts have been directed towards characterizing the effects of the microenvironment on cell behavior, e.g., in terms of proliferation, cell invasiveness and differentiation. However, it is interesting to note that SH-SY5Y cells have been successfully exploited for neurodegenerative studies, and in particular Parkinson’s and Alzheimer’s diseases, accounting for 14% of the articles. Indeed, they are usually chosen for their catecholaminergic (though not strictly dopaminergic) neuronal properties [71], which the cells manifest after treatment with retinoic acid (RA), a derivative of vitamin A and serum deprivation. It has also been demonstrated that SH-SY5Y cells possess two distinct populations, an N-type which differentiates to the neural lineage, and an S-type of substrate adherent cells which express characteristics of glial cells [72]. SH-SY5Y cells have been characterized as neurosteroid-producing cells expressing key steroidogenic enzymes, and were exploited for determining whether neurosteroidogenesis may be an endogenous mechanism involved in the protection against neurodegenerative processes [73]. Recently, Martin et al. [74] showed that this cell line can express glutaminergic markers when supplemented with B-27, which widens its field of application.

Once differentiated, SH-SY5Y cells show the formation of neural processes and functional synapses, as well as the production of neuron-specific enzymes, neurotransmitters and neurotransmitter receptors [68,75,76]. In addition, unlike other neural cell lines (e.g., the IMR-32), SH-SY5Y cells can develop a resting membrane potential and have been shown to possess voltage gated calcium channels upon differentiation. Moreover, the expression and localization of key molecules involved in the pathogenesis of Alzheimer’s disease has been shown to be dramatically altered in fully differentiated SH-SY5Y cells [77].

#### 3.1.3. Other Cancer Cell Lines

In addition to glioblastoma and neuroblastoma cell lines, 3D human brain models have been generated using cells originating from embryonal carcinoma and medulloblastoma. Human pluripotent embryonal carcinoma NTera2 (NT2) cells are widely used for in vitro neurotoxicity studies thanks to their ability to differentiate into post-mitotic neurons after treatment with RA [78,79]. NT2-laden spheroids differentiated with RA are known to express neural markers, such as tubulin and synaptophysin [80]. NT2 cells were also bioprinted, demonstrating their adhesion to fibrin gels [81].

The UW228-3 cell line was established from human posterior fossa medulloblastoma [82]. This line has been used in combination with human neural stem cells in ultra-low cell attachment plates for generating spheroids, which are exploited as an assay to test the effects of the cytotoxic drug etoposide. The spheroids comprise both cancer and stem cells, allowing the optimization of drug delivery for brain tumors in a more physiologically relevant model [83,84]. The Daoy cell line also derives from human medulloblastoma. Neurospheres laden with Daoy cells were cultured and compared with 2D monolayers grown on soft agar, revealing a higher expression of a protein related to cancer development in the 3D constructs [85].

### 3.2. Cell Lines Derived from Healthy Tissues

Although the majority of cell lines identified by the literature search had tumor origins, cell lines derived from healthy tissue can be used in 3D in vitro models of the human brain. Among these, the Lund human mesencephalic (LUHMES) line, which originates from the mesencephalon of a human subject, was established in 1998. The LHUMES neuron-like immortalized cells have been extensively characterized as a robust neuronal model suitable for neurodevelopmental studies, neurotoxicity and the modeling of brain diseases [86,87]. Indeed, they can be differentiated into dopaminergic neurons and thus are particularly suitable for modeling Parkinson’s disease in vitro.

Neural Progenitor Cells (NPCs) have a self-renewal capability and can give rise to healthy neural cell lineages. They can be obtained from iPSCs or derived directly from brain tissues [88,89]. In the context of this review, we considered the immortalized NPC lines because their reproducibility and proliferative potential can be exploited for the standardization of culture protocols for regulatory or preclinical tests. Commercial immortalized NPC lines employed in 3D neural in vitro models are ReNcell VM and ReNcell CX, derived, respectively, from the ventral mesencephalic and cortical region of the developing human brain and capable of differentiating into neurons and glial cells after the administration of growth factors. 

Table 5 summarizes the studies where the above-mentioned cell lines derived from healthy tissues were used for generating 3D in vitro models. 

## 4. Discussion

Over the last few decades, numerous studies have highlighted the superiority of 3D cultures with respect to monolayers as they are able to better recapitulate the morphology and architecture of tissues and cells in their native environment, both in physiological and diseases conditions. 3D systems based on human neural cell lines exhibit specific cellular and molecular features which occur in vivo [71,97] and support the expression of typical neural phenotypes and markers. Given their higher reproducibility with respect to primary cells, neural cell lines are fundamental for the definition of a standard brain environment useful for regulatory applications. 

Such 3D models can be realized using both scaffold-less and scaffold-based strategies [83,84]. The former typically exploits the predominance of cohesive forces with respect to adhesion forces when cells are cultured on low attachment plates or in suspension conditions (i.e., the hanging drop method or dynamic suspension culture with bioreactors and orbital shakers) [84]. On the other hand, the scaffold-based strategy is usually obtained through cell encapsulation in polymeric solutions which undergo gelation in cytocompatible conditions or through cell seeding and colonization of pre-formed scaffolds [98]. Scaffold-based constructs can be obtained either by casting or rapid prototyping methods (e.g., bioprinting) [99,100]. Among the materials used to replicate a tissue 3D matrix, alginate is widely used for its mild and rapid gelation in contact with aqueous solutions containing divalent ions [101,102], enabling the formation of spheroids with controlled shapes through the tuning of different bioprinting parameters, such as solution viscosity, extrusion speed and needle dimension [102,103].

Several examples of alginate-based spheroids laden with the cell lines described in this review can be found in the literature, in combination with different cell adhesive materials such as gelatin and collagen [36,57,61,64,70,93]. Interestingly, some tests have never been reported; for example there are no reports on SH-SY5Y differentiation in alginate-based spheroids, although studies have been carried out in other 3D gels fabricated with different methods and geometries. Some examples include collagen gels [68], collagen-coated nanocellulose [104] or silk scaffolds [53], hyaluronic acid scaffolds [66] and chitosan–graphene oxide nanocomposite hydrogels [65]. These methods, materials and geometries could enable the creation of more physiologically relevant models, because they allow cells to grow and connect to each other directly in 3D.

Although the results obtained with the generation of spheroids are promising, more systematic studies are still needed to exploit their versatility. For instance, 3D constructs may differ significantly in terms of mechanical and transport properties. This is mainly due to the high variability of the biomaterials used for fabricating them. Matrigel- and ECM-derived matrices are known to suffer from ‘batch syndrome’; on the other hand, alginate is more reproducible, but it is available in different molecular weights, and the protocols for obtaining spheroids involve different concentrations and crosslinking methods. Moreover, as alginate is a marine-plant-derived material, it lacks cell binding sites such as RGD motifs. For this reason, some attempts to combine alginate with cell adhesive materials, e.g., gelatin [58], can be found in the literature. The use of composite materials and biofabrication strategies is increasing as they enable finer control of the geometrical features of the scaffold and tuning of material mechanical properties [81], both of which are known to influence cell behavior [83,105,106]. This is even more important for brain tissue which has a low elastic modulus that increases with age (from around 110 Pa in neonate up to ≈1 kPa in adults [107]), and which changes significantly in some neuropathologies [108].

The applications of the 3D in vitro models described in Table 2, Table 4 and Table 5 can be divided into three main fields: oncology, neurodegenerative diseases, and neurotoxicity. Most of the studies described in this review employ 3D neural models for inspecting their responses against drugs, treatments, and chemotherapeutic agents (e.g., temozolomide, epigallocatechin gallate, natural killer cells, nanoparticles, and rotenone) [33,34,35,41,42,70,91,92]. For these studies, the 3D models represent an advanced tool more closely resembling the characteristics of in vivo tumor tissues. Indeed, the resistance of human tissues to anticancer drugs is a crucial factor which needs to be assessed and characterized for developing more efficient treatments. The delivery of such compounds should be investigated and optimized considering the whole microenvironment [109]. Furthermore, the possibility of tailoring some microenvironment features, e.g., matrix stiffness [37,38], for assessing the cell response to different conditions is supported by 3D models and represents one of their advantages over conventional monolayers. Finally, cell proliferation and invasion patterns can be studied in a more relevant context in 3D models [36,39,40,64,65,69].

The development of 3D in vitro models mimicking the in vivo cell environment and behavior is also essential to obtain relevant results in toxicity testing [68,110]. A number of studies compare cell responses to toxic compounds in 3D and 2D models developed with human neural cell lines. Differentiated neuroblastoma SH-SY5Y cells exhibit lower sensitivity to toxins when cultured in 3D constructs than in 2D ones [66,67]. Furthermore, 3D models allow the inspection of the influence of the micro-environment on the cell response and sensitivity to neurotoxins [84]. For example, matrix stiffness is responsible for regulating cell sensitivity to toxins, with softer matrices reducing cell viability [43]. Chemicals or materials used in therapy (e.g., gold nanoparticles) should also be assessed to reveal any adverse effects on cell physiology when administered in different concentrations. Three-dimensional models enable the assessment of cell responses and their capability for recovery after washing out the compound in a more physiological context than monolayers [91,92]. When generated from human neural cell lines, 3D models can be judiciously employed for the high throughput screening of neurotoxic compounds, to gain as much knowledge as possible about the potential adverse effects and risks of chemicals and drugs on cell viability [80,93,94].

A wide class of pathologies is included under the definition of neurodegenerative diseases, all of them characterized by the degeneration and death of neural cells [111,112]. The development of models closely resembling in vivo neural tissues is fundamental to advance our understanding of the pathogenesis and progress of neurodegenerative diseases, as well as to assess treatment efficacy. Most of the studies analyzed in this review regarding neurodegenerative diseases employ the neuroblastoma SH-SY5Y cell line to develop 3D models where cells are differentiated into mature neurons. The efforts are directed at creating more in vivo-like models to better understand features of neurodegenerative diseases [50,51,52,54,57], pathogenesis [53] and treatment effects [60,61]. Some investigations are focused on the optimization of the differentiation protocols, and assess the capability for obtaining differentiated cells with electrically active behavior [58,59,62,63,90]. Three-dimensional models can recapitulate the salient features of neurodegenerative diseases at the microscale and, with respect to 2D models, they better resemble features of in vivo tissues [55,56]. The use of NPC lines in 3D models allows the generation of a greater level of physiological relevance since different types of neural cells can be represented [95,96].

## 5. Conclusions and Future Perspectives

The ability to produce in vitro models with neural cells has been fundamental to advancing the understanding of the central nervous system’s (CNS) function at the microscale, as well as of the disease mechanisms underlying neuropathologies and neurotoxicity [14,113]. We argue that some of the challenges associated with culturing primary neural cells or stem cells could be overcome with the use of neural cell lines. Indeed, since these cells express human-specific proteins and have a complete human genomic profile, they have successfully been used for different applications, as reported in this review.

From our literature search, the first thing that stands out is the heterogeneity of the studies reported and the fact that many of them do not carry on a systematic characterization of the cell lines or materials involved. Long-term culture often leads to the accumulation of mutations in such cells, resulting in outcomes which are difficult to reproduce in laboratories [114]. We also underline that most of the human cell lines used in 3D neural models originate from tumors; hence, they may not recapitulate the properties of neural cells in vivo [14]. For example, they usually show a higher sensitivity toward oxidative stress with respect to primary cells [115]. Unless characterized and quantified, these factors may limit the translatability of the results, particularly when considering clinical applications.

The regulatory testing of chemical substances to define their limits of safety for humans and the environment using in vitro methods requires a high degree of inter-lab reproducibility and throughput, with tightly defined experimental protocols. This is necessary for compliance with the Guidance on Good Cell Culture Practice (GCCP) standards [116]. Risk and hazard assessments are then carried out, using a tiered approach which considers the integration of data from different endpoints and routes, rates, and duration of exposure. Safe doses are always estimated conservatively with built in precautions, which does not account for experimental variability [117]. Most regulatory authorities do agree that animal tests should be minimized and encourage the development of alternative or non-animal methods for chemical safety assessment. Although very few in vitro methods for chemical safety assessment have been approved by the Organization for Economic Co-operation and Development (OECD), the USA’s Environmental Protection Agency (EPA) and the European Chemical Agency (ECHA) to date, several are based on cell lines. This is partly because of their accessibility and reproducibility, as well as a great deal of investment in characterization and protocol development.

We suggest that the 3D culture of neural cell lines should be improved to promote their use in safety and toxicity testing of chemicals at a regulatory level, leveraging the versatility of cells such as SH-SY5Y to exploit their full potential. Further investigations should be performed towards the standardization of protocols and towards the identification of supplements able to generate different classes of neurons in a controlled manner. Alternatively, since glial cells modulate neuron function and signaling, while neurons generate and propagate electrical and chemical signals [118], neuroblastoma and glioblastoma cell lines could be co-cultured with the aim of delivering a more physiologically relevant model of the human brain. Additionally, 3D models based on NPC lines should be exploited for their ability to differentiate into the cell types that constitute the brain. In this way, we will be able to deliver a cellular model where both the main actors in the CNS—neurons and glia—are present. The future perspectives that we suggest for improving the development of more physiologically relevant 3D in vitro models of the human brain, leveraging neuronal cell lines, are summarized in Figure 2. Once standardized and characterized [119], their full-blown reproducibility can be exploited for generating large-scale studies based on 3D spheroids for high-throughput chemical safety and toxicity testing, as well as for oncological or neurodegenerative pre-clinical screening applications [75,120,121,122].

## Figures and Tables

**Figure 1 cells-12-01181-f001:**
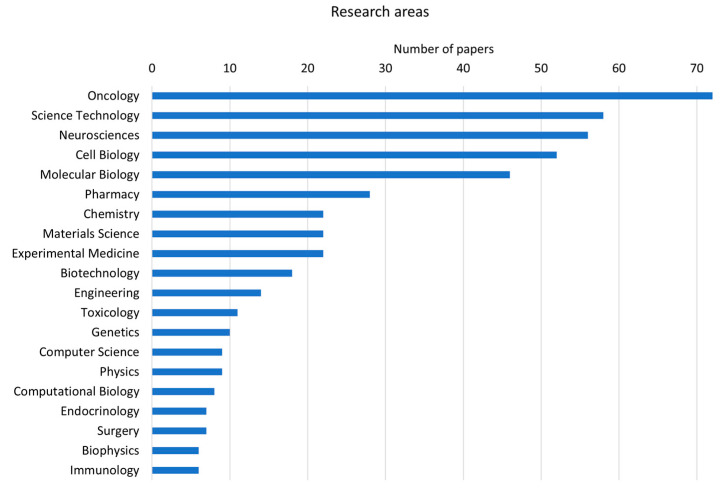
Bar chart of the first 20 research areas to which the papers retrieved from the literature search belong (data derived from Clarivate Web of Science, Copyright Clarivate 2022. All rights reserved).

**Figure 2 cells-12-01181-f002:**
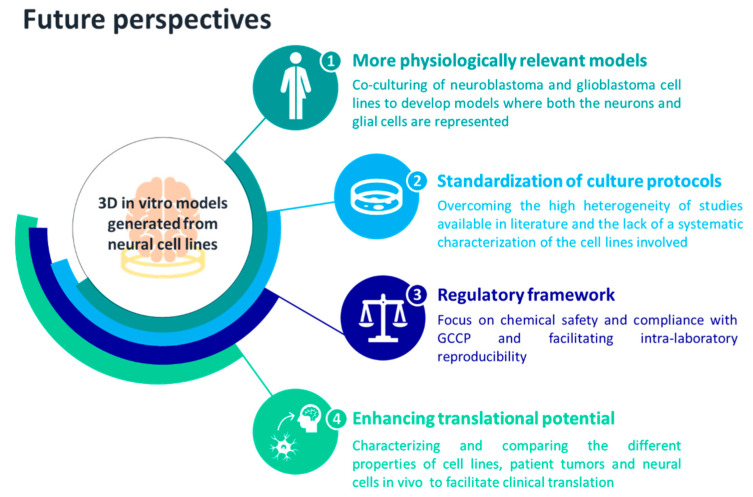
Future perspectives proposed for exploiting neural cell lines in 3D culture systems.

**Table 1 cells-12-01181-t001:** Characteristics of the most common glioblastoma cell lines and year first cited.

Cell Line	Origin	Gender and Age	Morphology	[Ref.], Year
U-87MG	Malignant glioma (likely glioblastoma)	Male, unspecified	Epithelial	[28], 1968
U-251MG	Glioblastoma-astrocytoma	Male, 75 years old	Pleomorphic/ astrocytoid	[29], 1984
U-373MG	Glioblastoma-astrocytoma	Male, 75 years old	Pleomorphic/astrocytoid	[30], 1989
T-98G	Glioblastoma multiforme	Male, 61 years old	Fibroblast	[31], 1979
A-172	Glioblastoma	Male, 53 years old	Fibroblast	[32], 1973

**Table 2 cells-12-01181-t002:** Three-dimensional models generated with glioblastoma cell lines.

Cell lines	Application	Materials and Methods	Main Findings	[Ref.], Year
U-87MG	Oncology	Self-assembled spheroids in agarose-coated 96-well plates treated with increasing concentrations of temozolomide	Spheroid growth influenced by administered dose	[33], 2015
		Self-assembled spheroids in agarose-coated 96-well plates treated with an inhibitor of the NOTCH signaling pathway	Reduced resistance of treated cells within spheroids to chemotherapeutic agents	[34], 2016
		Gene expression of spheroids obtained in low attachment wells compared with 2D controls	Upregulated gene expression of the inspected molecular characteristics in the 3D spheroid models compared with the 2D model	[35], 2021
		Self-assembled spheroids laden with wild-type and cells with increased malignancy implanted in collagen-I gels.	Differences in the cell proliferation between the wild-type and the more malignant ones due to lower cell adhesion	[36] 2007
		Spheroids with PEG-based hydrogel matrix with characteristics mimicking the physiological and glioblastoma-altered properties of in vivo ECM	Reduced cell proliferation and spreading on stiffer matrices	[37] 2014
		Bio-printed 3D constructs laden with glioblastoma and monocytic cells compared to 2D controls for cancer drug sensitivity	Optimization of the bio-printing procedure to promote a tumor microenvironment; 3D showed higher drug resistance than 2D	[38] 2020
		Co-culture of glioblastoma and endothelial-like cells in scaffolds fabricated with two-photon lithography, with microtubes resembling capillaries	Development of a realistic and 1:1 scale system mimicking the blood–brain barrier with good adhesion and covering by both cell types	[39] 2018
		Bioprinting of cell-laden 3D structures with a bioink made of fibrin, alginate and genipin	Good viability and tendency to form spheroids resulting in a more physiologically relevant glioblastoma model	[40] 2019
U-87, SHG-44 and U-251		Multicellular spheroids supplemented with B27, human basic fibroblast and epidermal growth factors, treated with EGCG for evaluating inhibition of cell stemness	Efficacy of the EGCG treatment in inhibiting cell viability and migration and inducing cell apoptosis, hence of potential in assessing glioblastoma therapy	[41] 2015
U-87MG, T-98G, A-172 and UW473		Compact multicellular spheroids formed with type-I collagen colloidal solutions (with increasing collagen concentration from 0 to 80 mg mL^−1^)	Development of a cheap and accessible method for building multicellular spheroids, usable for drug screening and glioblastoma cell infiltration	[42] 2022
U-87MG, U-251MG and IMR-32	Neurotoxicity	Spheroids obtained encapsulating cells in alginate, with concentration of 0.25 or 1% weight/volume and exposed to different toxins for 24 hr for testing cell viability	Higher sensitiveness to the toxins of the cells within the soft matrices than those in the stiffer ones, suggesting a role of matrix stiffness in neurotoxicity regulation	[43] 2014

PEG: poly(ethylene-glycol); ECM: extracellular matrix; EGCG: epigallocatechin gallate.

**Table 3 cells-12-01181-t003:** Characteristics of neuroblastoma-derived cell lines and year first cited.

Cell Line	Origin	Gender and Age	Morphology	[Ref.], Year
SH-SY5Y	Thrice cloned subline of the neuroblastoma cell line SK-N-SH	Female, 4 years old	Neuroblast	[46], 1973
IMR-32	Neuroblastoma	Male, 13 months old	Neuroblast; fibroblast	[47], 1970
HTLA-230	Neuroblastoma	Male, 11 months old	Round to bi-polar morphology	[48], 1992
Kelly	Neuroblastoma	Female, 1 year old	Round to fusiform with polar neurite processes	[49], 1982

**Table 4 cells-12-01181-t004:** Three-dimensional models involving neuroblastoma cell lines.

Cell lines	Application	Materials and Methods	Main Findings	[Ref.], Year
SH-SY5Y	Neurodegenerative diseases	Cells grown either on Matrigel or ECM scaffolds, differentiated with retinoic acid	3D models of the alpha-synuclein pathology associated with PD	[50,51,52], 2016, 2019, 2022
		RA-differentiated SH-SY5Y cells grown in silk-hydrogel or Matrigel, exposed to neurotoxicants	Model exploitable for studying the pathogenesis of PD	[53], 2022
		Cells grown on 3D nanoscaffold fabricated with polyacrylonitrile and Jeffamine^®^ doped polyacrylonitrile	Improved survival, growth and sensitivity to treatments mimicking PD features	[54], 2020
		Wild type and tau-mutated cells seeded on well plates, placed on a shaker to generate spheroids	Salient features of AD at the microscale recapitulated better by the spheroid model than 2D cultures	[55,56], 2010, 2012
		3D printed structures laden with cells in alginate and gelatin, using commercial printer	Good cell viability, maintenance of the 3D structure and spatial organization	[57], 2019
		Conductive and porous scaffolds fabricated by electro-polymerization using carbon nanotubes and PEDOT	Good biocompatibility shown by the improved tubulin expression on conductive scaffolds	[58], 2020
		Bacterial nanocellulose scaffolds coated with collagen I for promoting cell adhesion and differentiation	Functional action potentials were observed thanks to electrophysiological recordings	[59], 2013
		Collagen sponges (BIOPAD™) seeded with cells for investigating the neuroprotective effect of phytochemicals	Improved cell viability, upregulated antioxidant and insulin-degrading enzymes and reduced glutathione levels	[60], 2019
		0.3 % *w/v* alginate beads, obtained via syringe-pump-controlled extrusion from 15 to 27G needles, coated with 0.1% *w/v* poly-L-ornithine or 0.3% *w/v* hyaluronic acid	Suitability for CNS implantation and delivery of therapeutic cells for the treatment of neurodegenerative disorders	[61], 2022
		3D bioprinting of cells with bioinks composed of nanofibrils alginate and single-walled carbon nanotubes	Conductive scaffold-promoted cell differentiation (TUBB3 and NESTIN expression)	[62], 2020
		Cells seeded on scaffold generated by two-photon lithography of gelatin–methacryloyl and impregnated with magnetoelectric NPs	Electrostimulation allowed cell differentiation in the absence of chemical factors (neurite outgrowth with multipolar shape)	[63], 2020
	Oncology	Cells encapsulated in 2% *w/v* alginate thanks to electrohydrodynamic jetting and cultured for 4 weeks	Tissue maturation and higher cell viability, metabolic activity and proliferation level than cells cultured on TCP	[64], 2018
		Generation of chitosan (CH)–graphene oxide (GO) nanocomposite hydrogels seeded with cells	Cell differentiation (extensive neurite outgrowth) promoted by the CH–GO hydrogels	[65], 2021
	Neurotoxicity	3D hyaluronic acid-based hydro-scaffold (BIOMIMESYS^®^) seeded with cells	Higher neuronal differentiation and lower sensitivity to neurotoxic compounds with respect to 2D cultures	[66], 2021
		Microporous silk scaffolds coated with poly-L-ornithine and laminin, seeded with cells, encapsulated with collagen or Matrigel, and exposed to 1-methyl-4-phenylpyridinium	During differentiation, reduced proliferation and higher sensitivity to neurotoxins in comparison with 2D cultures	[67], 2020
		Cells encapsulated in 1 mg/mL collagen gels obtained by casting in Petri dishes and differentiated	Lower responsiveness of cells in 3D to potassium-induced cell depolarization with respect to 2D	[68], 2006
IMR-32, Kelly and SH-SY5Y	Oncology	Cells in collagen-based porous scaffold. Assessment of cell proliferation, viability and spatial within the scaffolds	Precise manipulation of cells and ECM components allowed by the 3D culture system; environment more physiologically similar to tumor tissue	[69], 2021
HTLA-230 and SH-SY5Y	Oncology	Cells suspended in alginate and manually extruded for mimicking the extracellular microenvironment experienced by tumor cells in in vivo settings	Reduced sensitivity to imatinib mesylate—a cytotoxic drug—with respect to cells cultured in monolayer and characteristics similar to the in vivo immunophenotype of tumor cells	[70], 2019

PD: Parkinson’s disease; AD: Alzheimer’s disease; PEDOT: poly(3,4-ethylenedioxythiophene); ECM: extracellular matrix; RA: retinoic acid; CNS: central nervous system; TCP: tissue culture plate; NP: nanoparticle.

**Table 5 cells-12-01181-t005:** Three-dimensional models with cell lines derived from healthy neural tissues.

Cell line	Application	Materials and Methods	Main Findings	[Ref.], Year
LUHMES	Neurodegenerative	3D constructs obtained by shaking (80 rpm) of cells seeded in 6-well plates with differentiation medium	Optimization of the differentiation protocol for a 3D construct with the formation of a pronounced neuronal network	[90], 2016
	Neurotoxicity	Spheroid formation by differentiation with neurotrophic factor and shaking (80 rpm). Treatment with different NP concentrations	Alteration of cell physiology and morphology of the spheroid surface provoked by the NPs, with induction of neurotoxic effects at the highest concentrations	[91], 2019
		3D constructs obtained by shaking (80 rpm) followed by 24 h exposure to rotenone	Recovery of ATP levels, mitochondria functions and neurite outgrowth after rotenone wash out showing good functional recovery	[92], 2018
ReNcell VM	Neurotoxicity	Cells encapsulated in alginate and Matrigel and bioprinted on microarray chip platforms	Successful establishment of miniaturized 3D culture of cells in alginate–Matrigel matrices useful for assessing toxicity	[93], 2018
		Microarray chip-based platform for the screening of the effect of 12 toxicants on neuronal differentiation	Enhanced neurogenesis and decreased astrocyte differentiation with the combined treatment of RA and CHIR	[94], 2019
	Neurodegenerative diseases, neurotoxicity	Direct write printing of a conductive polymer for the development of a 3D electrical stimulation tool of cells encapsulated within a conductive biogel	In situ differentiation of the NPCs into neurons and neuroglial cells and formation of tissue with high density and mature neurons	[95], 2019
ReNcell CX	Neurodegenerative diseases, neurotoxicity	Direct write printing of cells over a supporting polysaccharide (alginate, carboxymethyl-chitosan, and agarose)	In situ differentiation of NPCs to neurons with synaptic connections and spontaneous electrical activity	[96], 2016

NP: nanoparticle; RA: retinoic acid; CHIR: GSK3 inhibitor CHIR-99021; NPCs: neural progenitor cells.

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
