# Peer review of "Advanced 3D Models of Human Brain Tissue Using Neural Cell Lines: State-of-the-Art and Future Prospects"

_cells, 2023, doi:10.3390/cells12081181_

Round 1
Reviewer 1 Report
The manuscript entitled” Advanced 3D models of the human brain using neuronal cell lines: state of art and future prospects” by Rachele Fabbri and collaborators presents several 3D models for different neuropathologies using commercially available cell lines. The idea of the study is interesting. However, some concerns are listed below.
The title of the manuscript is misleading. The authors actually summarize 3D models of human brain pathologies, such as brain cancer or neurodegenerative disorders. Moreover, none of the listed cell lines are neuronal cell lines. It is true that some of the cell lines, such as SH-SY5Y, may present neuron-like features, but only after they are subjected to a differentiation process.
Also, considering the title, it would be expected that the authors will emphasize the advantages / disadvantages of each 3D model that they listed. However, this point has been partially covered in the discussion section.
Please check the tables. In table 2 and 4, the names of some cell lines are only partially written.
Author Response
Reviewer 1
The manuscript entitled” Advanced 3D models of the human brain using neuronal cell lines: state of art and future prospects” by Rachele Fabbri and collaborators presents several 3D models for different neuropathologies using commercially available cell lines. The idea of the study is interesting. However, some concerns are listed below.
The title of the manuscript is misleading. The authors actually summarize 3D models of human brain pathologies, such as brain cancer or neurodegenerative disorders. Moreover, none of the listed cell lines are neuronal cell lines. It is true that some of the cell lines, such as SH-SY5Y, may present neuron-like features, but only after they are subjected to a differentiation process.
Also, considering the title, it would be expected that the authors will emphasize the advantages/disadvantages of each 3D model that they listed. However, this point has been partially covered in the discussion section.
Thank you for appreciating our work and for the constructive comments. We have expanded the types of cell lines in the revised version. In particular, we added Daoy, LUHMES, ReNcell VM and ReNcell CX cell lines, which are not of tumor origin (lines 217 and following). Furthermore, we expanded the description of the studies where NT2 and UW228-3 are used (lines 181-196). Accordingly, the title of the manuscript was changed slightly to “Advanced 3D models of the human brain using neural cell lines: state of art and future prospects”. The new title reflects the fact that we have included all easily available neural cell lines and not just tumor derived neuronal lines.
In addition, to further emphasize the advantages and disadvantages of the 3D models described, we added a new Discussion section, where we summarized the main findings in terms of benefits of the use of 3D models with respect to conventional 2D monolayers, based on the information retrieved with the literature search and all the studies described in our manuscript. We underline that 3D models allow better models since they more closely recapitulate the microenvironmental features of in vivo tissues. Thus, they represent important tools for more precisely characterize the pathogenesis of neurodegenerative diseases and brain tumors as well as potential neurotoxic effects of drugs (lines 264-311).
Please check the tables. In table 2 and 4, the names of some cell lines are only partially written.
Thanks to the reviewer for this comment. We have fixed it and now the names of the cell lines are all completely written and readable in Table 2 (lines 135 and following) and Table 4 (lines 154 and following).
Reviewer 2 Report
Major concerns:
1. The title of this review is ‘Advanced 3D models of the human brain using neuronal cell lines: state of art and future prospects’, but this manuscript only discussed the 3D models of brain tumor cell lines. This view is too narrow. Thus, I would suggest that more 3D models using other neuronal cell lines, such as other immortal cell lines and iPSC cell line should be introduced in this review.
2. Although this review has included several commonly used 3D models of brain tumors such as self-assembled spheroids, multicellular spheroids, spheroids obtained encapsulating cells in alginate, and so on, some other 3D models like the 3D-printing models also need to be discussed.
3. It is expected to see some indications on the choice of cell-line derived models in research. Hence, it would be better if some summaries on current applications of cell-line-derived 3D models can be provided.
Minor concerns:
1. Brief introduction on how the 3D in vitro model is generated from neural cell lines. The material & method contained in the table is too simple.
2. I would suggest the year of publications added to the table so that the reader could see the advancement of the neural cell line 3D model in the past decades.
3. Figure 2 marks “3d in vitro model of the human brain”. It should be changed to “neural tumor cell line derived 3D in vitro model” based on the content.
Author Response
Reviewer 2
Major concerns:
The title of this review is ‘Advanced 3D models of the human brain using neuronal cell lines: state of art and future prospects’, but this manuscript only discussed the 3D models of brain tumor cell lines. This view is too narrow. Thus, I would suggest that more 3D models using other neuronal cell lines, such as other immortal cell lines and iPSC cell line should be introduced in this review.
We would like to thank the reviewer for his/her comment. We have expanded the types of cell lines in the revised version and better detailed the different origins of cell lines (lines 197-218). We added Daoy, LUHMES, ReNcell VM and ReNcell CX cell lines, which are not of tumor origin (lines 217 and following). Furthermore, we expanded the description of the studies where NT2 and UW228-3 are used (lines 181-196) and included two main sections for the description of the reports retrieved with the literature search: Section “3.1 Cancer cell lines” and Section “3.2 Cell lines derived from healthy tissues”. Thanks to these sections we are able to widen the scope of our work, according to the reviewer’s comment.
Accordingly, the title of the manuscript was changed slightly to “Advanced 3D models of the human brain using neural cell lines: state of art and future prospects”. The new title reflects the fact that we have included all easily available neural cell lines and not just tumor derived neuronal lines.
Although this review has included several commonly used 3D models of brain tumors such as self-assembled spheroids, multicellular spheroids, spheroids obtained encapsulating cells in alginate, and so on, some other 3D models like the 3D-printing models also need to be discussed.
We thank the reviewer for this comment. Indeed, some of the studies we describe also developed 3D printed in vitro models or constructs obtained by rapid prototyping techniques. Therefore, papers where 3D printed models are described are highlighted and we also added more references on this topic (Refs 40, 41, 42, 63, 64 in Tables 2 and 4). Furthermore, the Discussion section also remarks on 3D printing techniques for the development of these models (lines 228-263).
It is expected to see some indications on the choice of cell-line derived models in research. Hence, it would be better if some summaries on current applications of cell-line-derived 3D models can be provided.
We added a summary of the main fields of application of the cell-line derived 3D models in the Discussion section, to give an overview of the current use of these models in the oncology, neurodegenerative diseases and neurotoxicity fields (lines 264-311). We think that this has helped us to improve the quality and completeness of our work, thanks to the reviewer’s suggestion.
Minor concerns:
- Brief introduction on how the 3D in vitro model is generated from neural cell lines. The material & method contained in the table is too simple.
We added this description in the Discussion section (lines 228-263).
- I would suggest the year of publications added to the table so that the reader could see the advancement of the neural cell line 3D model in the past decades.
We thank the reviewer, we added the year of publication in the tables where the studies are described, to make reader aware of the advancement of neural cell line-based 3D models in the past few decades.
- Figure 2 marks “3d in vitro model of the human brain”. It should be changed to “neural tumor cell line derived 3D in vitro model” based on the content.
We changed figure 2 as suggested to “3D in vitro models generated from neural cell lines”.
Round 2
Reviewer 1 Report
The quality of the manuscript” Advanced 3D models of the human brain using neural cell lines: state of art and future prospects” by Rachele Fabbri and collaborators has been certainly improved by the authors. However, certain concerns still need to be addressed.
The title still does not fully represent the content of the manuscript. In general, 3D models of human brains refer to brain organoids and brain-on-a-chips. Please check and adjust.
Both Discussion and Conclusions sections are mainly focused only on SH-SY5Y cell line. Please check and adjust.
Author Response
Reviewer 1
The quality of the manuscript” Advanced 3D models of the human brain using neural cell lines: state of art and future prospects” by Rachele Fabbri and collaborators has been certainly improved by the authors. However, certain concerns still need to be addressed.
The title still does not fully represent the content of the manuscript. In general, 3D models of human brains refer to brain organoids and brain-on-a-chips. Please check and adjust.
Both Discussion and Conclusions sections are mainly focused only on SH-SY5Y cell line. Please check and adjust.
We thank the reviewer for appreciating the work and for the suggestions. We have changed the title of the manuscript to “Advanced 3D models of the human brain tissue using neural cell lines: state of art and future prospects” for better representing the content of our manuscript according to the reviewer’s comment. Indeed, here we refer to 3D models as tools for replicating the tridimensional nature of brain tissue at a cell level.
We also checked and adjusted the Discussion section to make the considerations more general and focused on all the cell lines described in the paper (lines 236-245).
Reviewer 2 Report
The Authors have addressed all of my concerns. The revised manuscript is ready for publication.
Author Response
Reviewer 2
The Authors have addressed all of my concerns. The revised manuscript is ready for publication.
We thank the reviewer for this comment.